

# The evaluation of university management performance using the CS-RBM algorithm

Huifang Guo

Zhengzhou Vocational College of Finance and Taxation, Zhengzhou, Henan, China

## ABSTRACT

Amidst the ongoing higher education reforms in China, the escalated investments in colleges and universities underscore the need for an effective assessment of their performance to ensure sustainable development. However, traditional evaluation methods have proven time-consuming and labor-intensive. In response, a novel approach called CS-RBM (Crow Search Restricted Boltzmann Machine) prediction algorithm has been proposed for the educational management of these institutions. By integrating the CS algorithm and an enhanced RBM algorithm, this method facilitates the scoring of project performance indicators, bolstered by insights from user evaluation form reports. The comprehensive project performance is ultimately derived from this combination. Comparative analysis with the standard particle swarm optimization algorithm on public data sets demonstrates a remarkable 45.6% reduction in prediction errors and an impressive 34.7% increase in iteration speed using the CS-RBM algorithm. The accuracy of the tested data set surpasses 98%, validating the efficacy of the CS-RBM algorithm in achieving precise predictions and effective assessments. Consequently, this innovative approach exhibits promising potential for expediting and enhancing the performance evaluation of colleges and universities, contributing significantly to their sustainable development.

## INTRODUCTION

Education management is the key to promoting the use and benefit of university project funds and the key factor (*Guan-yuan, 2021*) in influencing the rational allocation of project funds. In recent years, China's institutions of higher learning have developed rapidly. According to the statistics of the Ministry of Education, by 2020 (*Juan, 2020*), a total of 3,005 institutions of higher learning nationwide have implemented the educational management of project funds in order to meet the large-scale requirements of finance and education departments. Only a small number of universities have completed the evaluation work. The functional department only considers whether the project work plan can be completed, which causes many problems (*Chen, 2019*). The performance target has deviated because the efficiency of project funds is rarely considered. The performance management system is not perfect, the index setting is not reasonable, and the index system weight standard is inconsistent, leading to a lack of educational performance management. When allocating funds to each project, performance management is often unreasonable. These include the allocation of funds according to experience, the separation

Corresponding author
Huifang Guo,
12004001@zzcsjr.edu.cn

of responsibilities, the lack of performance monitoring, the overemphasis on investment, the neglect of management, and the emphasis on expenditure. However, the evaluation of expenditure is lacking. In addition to the above problems, the problem of different weighting standards also leads to the inaccurate management of education funds, which easily exceeds each project's budget. Since only the past situation is used as a reference, most funds are allocated based on experience. Although these problems have been paid attention to, effective performance monitoring is often difficult to implement due to the current manual performance evaluation limitations. The problem of university performance has become the main problem of colleges and universities. How to reflect the budget income and performance output of education and realize the precise management of education? It is increasingly the focus of important issues. From a sustainable development perspective, how to effectively apply the information index has become an important solution.

Due to little consideration of project funds' efficiency, it deviates from the performance target. The performance management system is imperfect, and there are unreasonable indicator settings and different weight standards in the indicator system, resulting in unreal education management. In allocating funds for each project, the budget of more than 1 year is for reference (*Zighan & El-Qasem, 2021*). Such phenomena include allocating funds according to experience, separation of rights, lack of performance monitoring, excessive emphasis on investment, neglect of management, management, emphasis on expenditure, and neglect of assessment.

In addition to the above problems, the problem of different weight standards also leads to inaccurate education management, above the budget of each project. The 1-year amount is a reference, and some funds are allocated according to experience, separation of rights and responsibilities, non-implementation of performance monitoring, and excessive attention to investment. However, they usually do not pay attention to management, light management, heavy expenditure, and light assessment.

The problem of university performance has become a major issue in colleges and universities; how to embody the budget revenue and performance output and realize accurate education management. It has increasingly become the focus of important problems. From the perspective of sustainable development, the application of the Informatization index has become an important solution (*Zighan & El-Qasem, 2021*).

In this respect, the use of a swarm intelligence algorithm is very important for the implementation of system construction and has a large number of applications in various fields. Currently, the commonly used education management method in the field of scientific and technological innovation in colleges and universities is proposed in the data Envelopment analysis method (DEA) (*Yuning, 2018*). *Johnes (2006)* used the DEA model to evaluate the performance of scientific and technological innovation activities of 100 British universities, and the results showed that British universities obtained high technical efficiency and scale efficiency. *Wei & Zhong (2021)* gave an information education management system on personnel training, scientific research, management, and service in colleges and universities. *Moullin (2017)* believed that most education sectors faced two

main problems: first, improving the evaluation results of service users and stakeholders, and second, not increasing the overall cost.

In 2021, *Song-Shan et al.* provided the education management method of informatization in practice. However, the system depends on the traditional method, which predicts that results do not meet the requirements. Improving the traditional method and finding the evaluation index system is one of the keys to improvement, and constructing the achievement has become an important basis and guidance work. Such as building an evaluation system and adopting the genetic algorithm and particle swarm optimization (PSO) combined with the BP neural network algorithm to evaluate the quality of the information system (*Xiaohua & Chengxiang, 2020*). *Wang (2020)* established evaluation indexes for the operation and maintenance quality of the information system of power supply companies.

The literature review provides several potential directions for further innovative research in the field of university performance evaluation and education management. Firstly, building upon the emphasis on the Informatization index as a solution for accurate education management, it is essential to integrate swarm intelligence algorithms (such as CS-RBM) to enhance the effectiveness of system construction and improve overall performance evaluation. Secondly, conducting a comparative analysis between traditional methods (*e.g.*, DEA) and the aforementioned CS-RBM algorithm will help evaluate their respective strengths and weaknesses in assessing university performance. This analysis will provide valuable insights into the suitability of each method for specific evaluation scenarios. A third area of exploration involves constructing a comprehensive and multi-dimensional evaluation system that incorporates not only budget revenue and performance output but also factors such as sustainability, social impact, and student satisfaction. This holistic approach will yield a more comprehensive and well-rounded assessment of university performance. Furthermore, investigating new methods to build an optimal evaluation index system is crucial, encompassing various factors contributing to college and university performance. Integrating advanced techniques like genetic algorithms, particle swarm optimization, and other innovative approaches will facilitate the identification of relevant and impactful indicators. A fifth direction entails applying the CS-RBM algorithm and the refined evaluation index system to real-world contexts in colleges and universities, conducting case studies to demonstrate their effectiveness in diverse institutional settings, and providing practical recommendations for education management enhancement.

Based on the above research, this article proposes the CS-RBM algorithm based on the crow search algorithm and improved BP neural network. The main contributions of this article include the following three aspects:

1) The RBM network is improved, and it is proved that the improved RBM network has higher accuracy than the traditional deep learning method.
2) A CS-RBM algorithm combining the crow search algorithm and RBM network is proposed, which achieves more accurate prediction and faster iteration speed on public datasets WineQuality and Iris.

3) Through the analysis of the university performance data set and prediction results in this article, the university performance index system this article is established and verified. The verification shows that the prediction results have 98% similarity with the evaluation results.

## RELATED WORKS

Conducting a longitudinal study using the CS-RBM algorithm and the improved evaluation system will enable the monitoring of university performance over time, shedding light on the sustainability and long-term effects of management strategies. Lastly, incorporating stakeholder perception analysis into the evaluation process will be valuable. Investigating how stakeholders perceive university performance and combining their perspectives with the outcomes obtained through the CS-RBM algorithm will offer a deeper understanding of university performance from different viewpoints. By exploring these research directions, we can advance the field of education management, offering valuable theoretical and practical guidance for colleges and universities striving for sustainable development and accurate performance assessment.

### Construction of performance management index system

Performance management is based on analyzing the disadvantages of traditional education management. Performance management is an activation process in which all members of an organization jointly improve the performance of the organization through spiritual or material incentives to achieve preset goals (*Kosel, Wolter & Seidel, 2021*). Particularity of the performance indicator system of universities, to establish a scientific and reasonable education management system, which includes: defining the project tasks and objectives, objectively analyzing the content of the project tasks and objectives, distinguishing the priority of the tasks, and evaluating the projects, and finally obtaining the performance indicators (*Ha, Jung & Koh, 2021*). This article establishes the corresponding evaluation system block diagram according to the statistical law and system requirements. The framework of the evaluation system is shown in Fig. 1.

The construction of second-level indicators in this article depends on the specific situation. Different education management studies of information projects have different understandings of the definition of satisfaction, as shown in Table 1.

### RBMs model

The basic building blocks of a DBN are restricted Boltzmann machines (RBMs). An RBM is a type of energy-based probabilistic graphical model that consists of two layers: a visible layer representing the input data and a hidden layer capturing latent features. These layers are connected by symmetric weighted connections, meaning there are no connections between nodes within the same layer. This characteristic allows for efficient training using a technique called contrastive divergence.

DBNs are particularly effective at learning hierarchical representations, where lower layers capture simple patterns and higher layers capture more complex and abstract features. This hierarchical structure allows DBNs to excel in tasks like image recognition,

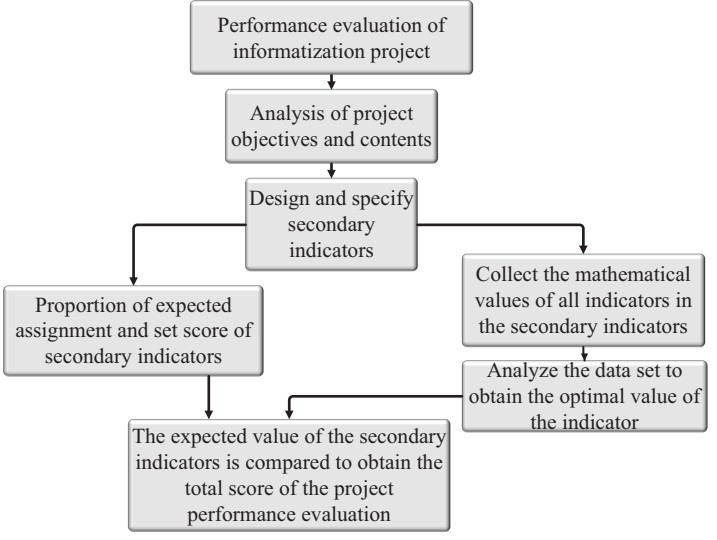

**Figure 1** **Evaluation system framework.**

speech processing, and natural language understanding. Once the individual RBMs are pretrained, the whole DBN is fine-tuned using traditional supervised learning techniques like backpropagation. During fine-tuning, the model learns to adjust its parameters to better fit the desired output labels.

After defining the evaluation criteria, this article improves the Boltzmann machine (RBMs) model, and the model structure is shown in Fig. 2.

In Fig. 2, the curve box represents the structure of RBMs, which is composed of multiple RBMs to form a modular feature extractor, where RBMs(i) is the RBMs, $i = 1, 2...L$. If the input feature is set to $(x_1, x_2,... x_n)$. According to the sample, characteristics can be divided into L feature modules, regardless of the order, the characteristics of the module to $(x_1, x_2, ... x_n)$, $(x_{n+1},...x_{n+2}) \rightarrow (x_{nL+1}...x_{nL})$ the characteristics of the module corresponds to the input layer of RBM. That layer of RBM is the first matter visible layer; the output of the corresponding hidden layer of the last RBM is the new feature data. Then the output of each RBM is reconstructed and used as the input of the top-level (generalized associative memory) classifier. The solid line box in the Figure shows a network structure of the top-level classifier, where yj (j = 1, 2,..., n) represents the network output value of the classifier. In this article, the deep learning model combines multiple RBMs feature extractors and top-level classifiers first to use RBMs to extract features from the sample data of different feature modules and then use top-level classifiers for classification processing. Add a classifier at the top level of the deep learning model, and college education management is a multi-category variable (*Ha, Jung & Koh, 2021*).

Therefore, this article uses the Softmax classifier as the top-level classifier. Softmax classifier is a generalization of the Logistic regression model for multi-classification problems, where the classification label y can take two or more values. Softmax classifier is a nonlinear model used for classification (*Hare Megan et al., 2021*). It can get high classification accuracy when combined with a deep neural network. The training process of the Softmax classifier is to obtain the optimal parameters of the model by iterating the

**Table 1  Performance satisfaction of informatization project.**

| Level I indicators | Level II indicators | Indicator description |
|---|---|---|
| Customer satisfaction | Satisfaction with system stability | System stability evaluation |
| | Satisfaction with system convenience | System convenience evaluation |
| | System convenience and function satisfaction | System convenience and function satisfaction evaluation |
| | Satisfaction of system security defense | Satisfaction evaluation of system security defense |

gradient of parameters according to the cost function. The value of S is calculated as shown in Eq. (1).

$$S = \left\{ \left( X^{(1)}, y^{(1)} \right), \left( X^{(2)}, y^{(2)} \right), \ldots, \left( X^{(N)}, y^{(N)} \right) \right\} \tag{1}$$

where $X^{(i)}$ is the sample, $Y^{(i)}$ is the corresponding category output value, $X^{(i)}$ and N is the number of samples. If there are K categories, then $y^{(i)} \in \{1, 2, \ldots, K\}$. We adopt the method of hypothesis function to analyze the probability value $\phi_j = p(y = j|X)$ that each output value belongs to a different classification and define the hypothesis function form as follows:

$$h_\theta\left( X^{(i)} \right) = \begin{bmatrix} p\left( y^{(i)} = 1 | X^{(i)}; \theta \right) \\ p\left( y^{(i)} = 2 | X^{(i)}; \theta \right) \\ \vdots \\ p\left( y^{(i)} = K | X^{(i)}; \theta \right) \end{bmatrix} = \frac{1}{\sum_{j=1}^{K} e^{\theta_j^T x^{(i)}}} \begin{bmatrix} e^{\theta_1^T X^{(i)}} \\ e^{\theta_2^T X^{(i)}} \\ \vdots \\ e^{\theta_K^T X^{(i)}} \end{bmatrix} \tag{2}$$

where $\theta = [\theta_1, \theta_2, \ldots, \theta_K]^T$ are the model parameters of the Softmax classified $p\left( y^{(i)} = j | X^{(i)} \right)$ notes the probability value that the sample $X^{(i)}$ belongs to the j-th class, and $\sum_{i=1}^{K} e^{\theta_j^T x^{(i)}}$ which normalizes the model output values so that their output values sum to one? The cost function of the Softmax classifier is in Eq. (3):

$$C(\theta) = -\frac{1}{N} \left( \sum_{i=1}^{N} \sum_{j=1}^{K} I\{y^{(i)} = j\} \log \frac{e^{\theta_j^T X^{(i)}}}{\sum_{s=1}^{K} e^{\theta_s^T X^{(i)}}} \right). \tag{3}$$

I (expression) is the indicator function, defined as follows: I am 1 when the expression is true and 0 when the expression is false. The partial derivative of Eq. (3) is obtained to obtain the gradient Eq. (4):

$$\nabla_{\theta_j} C(\theta) = -\frac{1}{N} \sum_{i=1}^{N} \left[ X^{(i)} \left( I\{y^{(i)} == j\} - p\left( y^{(i)} = j | X^{(i)}; \theta \right) \right) \right]. \tag{4}$$

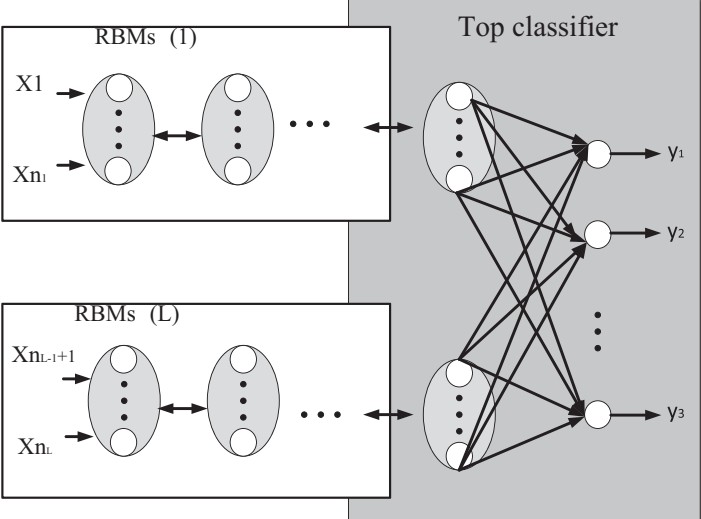

**Figure 2 Schematic diagram of modular deep learning model based on RBMs model.**

According to the structural characteristics of the sample data of university education management, the input features are divided into two feature modules. Set sample data is shown in Eq. (5):

$$S = \left\{ \left( x^{(1)}, \tilde{x}^{(1)}, y^{(1)} \right), \left( x^{(2)}, \bar{x}^{(2)}, y^{(2)} \right), \dots, \left( x^{(N)}, \tilde{x}^{(N)}, y^{(N)} \right) \right\}. \tag{5}$$

In Eq. (5), $x^{(i)} \in R^{n_1}, \bar{x}^{(i)} \in R^{n_2} \ y^{(i)} \in \{1, \dots, K\}$, $n_1 + n_2 = n$. N is the number of samples, and K is the number of categories representing the total number of input features. Moreover, x corresponds to the input features of RBMs(1) and RBMs(2), respectively, written in the following vector form in Eq. (6):

$$\begin{aligned} x &= [x_1, x_2, \dots, x_{n_1}]^T \\ \tilde{x} &= [\tilde{x}_1, \hat{x}_2, \dots, \hat{x}_{n_2}]^T \end{aligned} \tag{6}$$

Assuming that RBMs(1) and RBMs(2) have L1 and L2 layer networks, and the number of hidden layers of RBMs(1) and RBMs(2) is L1 and L2. The Greedy layer-by-layer unsupervised parameter training strategy and RBM training algorithm are used to initialize the connection weights and bias of RBMs in each layer. The parameters of the Softmax classifier are iteratively trained again, and the initial network parameters are obtained as follows: $(W^1, W^2, \dots, W^{4-1}), (V^1, V^2, \dots, V^{L_2-1})$. Let the number of hidden layer nodes of RBMs(1) and RBMs(2) be as follows: $(p_1, p_2, \dots, p_{L_1}), (q_1, q_2, \dots, q_{L_2})$ then there is $W^i \in R^{p_{i+1} \times (p_i+1)}, V^j \in R^{q_{j+1} \times (q_j+1)}$ network parameters are the matrix of bias and connection weights together. The propagation structure of RBMs(1) and RBMs(2) networks is shown in Fig. 3.

According to the forward propagation of the input feature data, the input state function of the j-th neuron in the l-th layer of RBMs(1) is $z_j^l, l = 2, \dots, L_1, j = 1, \dots, p_l$ after the

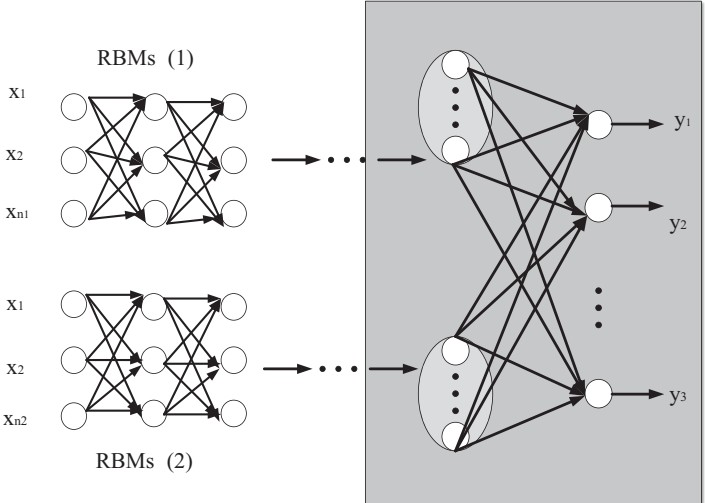

**Figure 3 Network propagation structure diagram of RBMs(1) and RBMs(2).**

activation function is applied, the output is $j_n$. Similarly, the input state function of the j-th neuron in layer l in RBMs(2) is $z_j^l, l = 2, \ldots, L_2, j = 1, \ldots, q_l$ as follows:

$$
\begin{aligned}
z^{l+1} &= W^l u^l, \quad u^l = \sigma(z^l) \\
z^{l+1} &= V^l \tilde{u}^l, \quad \tilde{u}^{-1} = \sigma(z^l)
\end{aligned}
\tag{7}
$$

where $w_{i0}^l = b_i^l, b_i^l$ denotes the bias of the node of the hidden layer of the RBM in RBMs(1), and also $v_{i0}^l = \tilde{b}_i^l; \quad u^1 = x, \hat{u}^1 = \hat{x}$ of RBMs(2) which means that $u^1$ and $\hat{u}^1$ are input feature vectors and $\sigma(x)$ is the Sigmoid function.

Each RBMs output reconstruction characteristic vector for $U = \left[ (u^4)^T, (\hat{u}^{L_2})^T \right]^T$, equivalent to the original input characteristics, $X = [x^T, \bar{x}^T]^T$ through a series of features transform to extract the new U, X to U is a mapping relation. $(U^{(i)}, i = 1, \ldots, N)$ are the input sample data of Softmax classifier. According to the assumption function of Eq. (6), the network output value of Softmax classifier can be obtained as shown in Eqs. (8) and (9).

$$
h_\theta \left( U^{(i)} \right) = \left[ \phi_1 \left( X^{(i)} \right), \phi_2 \left( X^{(i)} \right), \ldots, \phi_K \left( X^{(i)} \right) \right]^T
\tag{8}
$$

$$
\theta = [\theta_1, \theta_2, \ldots, \theta_K]^T, \quad U^{(i)} \leftarrow \left[ 1, \left( U^{(i)} \right)^T \right]^T
\tag{9}
$$

Among them $\theta = [\theta_1, \theta_2, \ldots, \theta_K]^T, \quad U^{(i)} \leftarrow \left[ 1, \left( U^{(i)} \right)^T \right]^T$

The loss function of the model is constructed by using the regularization method as shown in Eq. (10):

$$
C(\Theta) = -\frac{1}{N} \left( \sum_{i=1}^{N} \sum_{j=1}^{K} I\{y^{(i)} == j\} \log \phi_j \left( X^{(i)} \right) \right) + \frac{\lambda}{2} (\|\Theta\|_2)^2
\tag{10}
$$

where 0 is the regularization coefficient, the term is used to reduce the risk of being overfitted. It is the set of all parameters of the network as shown in Eq. (11):

$$W = \left(W^1, \ldots, W^{4-1}\right), V = \left(V^1, \ldots, V^{L_2-1}\right) \tag{11}$$

# CONSTRUCTION OF UNIVERSITY PERFORMANCE INDEX EVALUATION SYSTEM

## CS-RBM algorithm model

The CS-RBM algorithm in this article is based on the vast improvement of the crow search algorithm and RBM algorithm. The crow search algorithm is an algorithm proposed by Iranian scientists in 2016. The strategy of the crow search algorithm originates from the research on the crow population. Therefore, the main optimization strategy can be obtained according to its communication flow. Crow search algorithm has two main advantages: (1) The algorithm is non-greedy, so the iteration speed is fast. (2) By adjusting the perception function parameter value AP and other parameter values and weight values, the crow population search algorithm can focus on global search or local search ability, which is convenient for engineering use and adjustment. Therefore, the crow search algorithm has recently been widely applied in engineering and has low computational power requirements (*Agüero et al., 2021*). The Crow search algorithm can show strong adaptive ability. It can adjust the search parameters, improve the iteration effect by adaptive weights, and improve the population ability by Cauchy mutation, which has strong development potential. The idea behind the crow search algorithm is that members of the crow population can remember where other members are hiding food, steal other members of the crow population by tracking, and move their food hiding places. Then the crow algorithm search state is as follows:

$$X^{i+1} = \begin{cases} X^i + r_i \times fl^i \times (m^j - X^i) & r_i \geq AP^j, i \neq j \\ \text{a Random Position} \end{cases} \tag{12}$$

The flow of Crow algorithm is as follows:
1. The crow population number K is determined
2. Determine the population fitness L
3. Initialize data values
4. While i < imax
for i = 1: N
Define perceptual probability AP
if r > AP
Xi = $X^i + r_i \times fl^i \times (m^j - X^i)$
Else Xi = a Random Position
end if
end for
5. Get the data point fitness value
6. Update data value

7. end

The Restricted Boltzmann Machine (RBM) neural network possesses the capability of feedback regulation, which entails significant time and space consumption due to iterative feedback required for weight and threshold updates. To accelerate the convergence of the RBM neural network and optimize network weights and thresholds, two aspects are considered: inertia weight and learning factors. The crow search algorithm is employed to initialize the values of neural network weights and thresholds. Through a study of gradient descent, a prediction model is established by training the model with sample data. The validity of the model is then tested using test sample data. By employing the crow search algorithm and optimizing the gradient descent process, the RBM neural network can achieve faster convergence and better performance in training and testing tasks. This optimization allows for more efficient updates of network weights and thresholds, contributing to the effectiveness of the prediction model in various applications. The structure of the CS-RBM algorithm model after sorting is shown in Fig. 4.

## Improvement and implementation of CS-RBM algorithm

RBMS, in this article, is a two-layer deep learning model. The learning algorithm of the model in this article is reverse fine-tuning, which is a supervised learning of the entire network. That aims to biologically reduce the random learning characteristics of network blindness and achieve better learning effects. The parameters in forward propagation from the input layer to the top layer of the network are obtained by coarse-tuning learning, that is, unsupervised training. At the same time, fine-tuning is supervised learning to update the network parameters according to the minimization loss function. The regularization term of the loss function, the specific gradient formula fine-tuned according to the traditional BP algorithm, is given below.

Let the error of the $J$ neuron in the top layer be j, the error of the j-th in the L layer in RBMs(1) is $\delta_j^l$, and the error of the $J$ neuron in the $L$ layer in RBMs(2) is $\bar{\delta}_j^l$, the value of $\bar{\delta}_j^l$ and $\delta_j^l$ is shown is Eq. (13):

$$\xi_j = \frac{\partial C}{\partial o_j}, \delta_j^l = \frac{\partial C}{\partial z_j^l}, \bar{\delta}_j^l = \frac{\partial C}{\partial z_j^l} \tag{13}$$

$o_j = \theta_j^T U$ is the input value at the top level. Top layer neuron error is

$\xi = [\xi_1, \xi_2, \ldots, \xi_K]^T$

We can get the value of $\delta_j^l$ is shown is Eq. (14):

$$\xi_j = -\frac{1}{N} \sum_{i=1}^{N} \left( I\{y^{(i)} == j\} - \phi_j \right). \tag{14}$$

The gradient formula of network parameters is obtained as shown is Eq. (15):

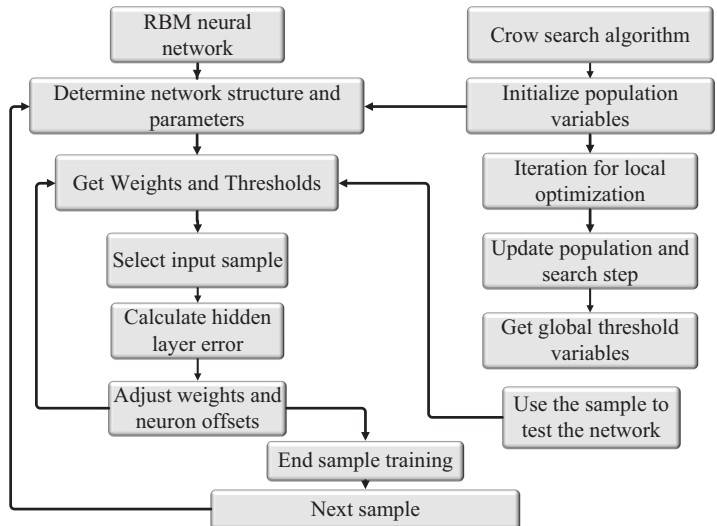

**Figure 4 Structure diagram of CS-RBM algorithm model.** The crow search algorithm for neural network weights and threshold as a matter of initialization value, through the study of gradient descent, the prediction model was formed by the training sample study by testing the validity of the model test sample data.

$$\frac{\partial C}{\partial \theta_j} = -\frac{1}{N} \sum_{i=1}^{N} \left[ U^{(i)} \left( I\left\{ y^{(i)} == j \right\} - p\left( y^{(i)} = j | U^{(i)}; \theta \right) \right) \right] + \lambda \theta_j$$

$$\frac{\partial C}{\partial w_{ij}^l} = \delta_i^{l+1} u_j^l + \lambda w_{ij}^l, \quad \frac{\partial C}{\partial v_{ij}^l} = \hat{\delta}_i^{l+1} \tilde{u}_j^l + \lambda v_{ij}^l (j > 0) \tag{15}$$

$$\frac{\partial C}{\partial w_{i0}^l} = \delta_i^{l+1}, \frac{\partial C}{\partial v_{i0}^l} = \hat{\delta}_i^{l+1}$$

Based on the above contents and the education management data, the specific implementation steps of the model based on RBMs are shown in Fig. 5. The detailed steps are described as follows:

Step 1: Divide the input feature data S into two feature module data sets: $I_1$ and $I_2$.

Step 2: The RBM training algorithm is used to obtain the output value of RBM of the first layer $u^2, \tilde{u}^2$ and connection weight $W^1, V^1$ respectively, according to the input data $I_1$ and $I_2$.

Step 3: Taking $u^2, \tilde{u}^2$ as the input data of the second RBM, based on the greedy layer-by-layer unsupervised pre-trained strategy and RBM training algorithm, the output values of each hidden layer $u^3, \ldots, u^4$ and $\hat{u}^3, \ldots, \hat{u}^{L_2}$ are iteratively obtained. The connection weight is $W^2, \ldots, W^{L_1-1} V^2, \ldots, V^{L_2-1}$, which is used as the initial network parameter.

Step 4: The output $u^{L1}, \tilde{u}^{L2}$ of the last RBM is reconstructed as the input data of the Softmax classifier, and the expected output data is used to train the classifier to obtain the initial parameters.

Step 5: Connect the pre-trained RBMs(1), RBMs(2) and Softmax classifier to form a deep learning network model with a classification function.

Step 6: Apply the BP algorithm to fine-tune the network.

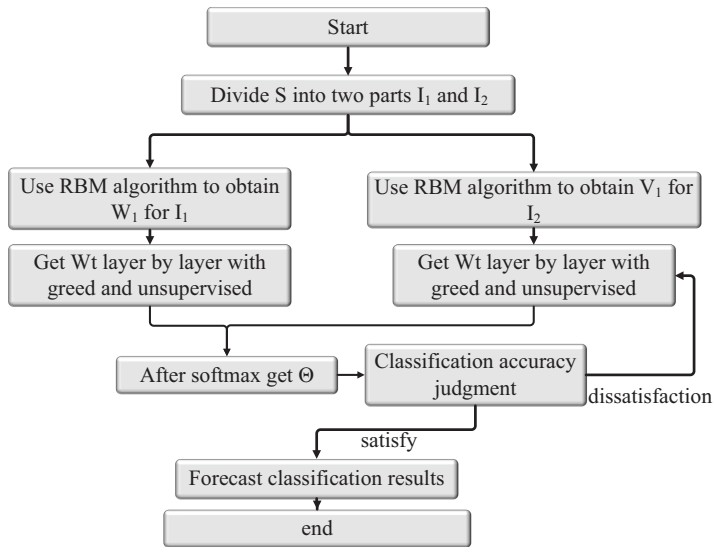

**Figure 5  Structure of RBM algorithm.**     

Step 7: If the classification accuracy exceeds the preset value or reaches the maximum number of iterations, end the learning; otherwise, go to Step 6. The structure of the obtained RBM algorithm is shown in Fig. 5.

## EXPERIMENT

### Experimental environment and data sets

The Iris dataset is a commonly dataset, also known as the Iris flower dataset, which is a kind of multivariate analysis dataset. The dataset contains 150 data records divided into three categories, with 50 data records. Due to the length, only part of the data samples is listed in Table 2. In the simulation experiment of this algorithm, the Iris dataset is divided into two groups. The dataset is divided into training and testing at a ratio of 7:3 into training and test. They used different samples in each group and flowers for 75 and 25. One group is a training sample, and the other is used as a test sample, and all data are normalized between [0,1].

WineQuality data set, there are two data subsets, one is Redwine, and the other is White wine. Redwine has 1,599 data records, and white wine has 4,898 data records; both data sets have 11 input features, and all features are continuous data; all data are normalized between [0,1]. The output is subjective ratings of the quality of wines made by professional tasters, ranging from 0 to 10.0 means that the quality of the wine is very bad, and 10 points mean that the quality of the wine is very good. The data sample is shown in Table 3. That can be used to analogy the education management process and verify the subjective value evaluation problem in education management. The inequality dataset is divided into two groups in this algorithm simulation experiment. The two data sets are divided into training data and test data with a ratio of 7:3, and each group has 750 sample data for training samples and 250 sample data for test samples.

**Table 2 Sample data table of IRIS dataset.**

| Sepal length (cm) | Sepal width (cm) | Petal length (cm) | Petal width (cm) | Species |
|---|---|---|---|---|
| 1 | 5.1 | 3.5 | 1.4 | 0.2 |
| 2 | 4.9 | 3.0 | 1.4 | 0.2 |
| 3 | 4.7 | 3.2 | 1.3 | 0.2 |
| 4 | 4.6 | 3.1 | 1.5 | 0.2 |
| 5 | 5.0 | 3.6 | 1.4 | 0.2 |

## Effectiveness analysis of improved RBM network

In order to verify the effectiveness of the RBM network, all experiments are based on the MATLABR2011b operation platform. In this section, we study the performance changes of the Softmax classifier after feature extraction from input data with the increase of the new feature expression dimension using RBM and PCA. Input features and output features are taken as input features together, the dimension of input features is 56, and the dimension of output variables is 4. When the dimension of new features increases from 1 to 56, the changes of PCA, RBM-expression and RBM fine tuning on the performance of the Softmax classifier are recorded. PCA-expression and RBM-expression represent the classification method of feature data after PCA and RBM feature extraction by Softmax classifier. RBM-fine-tuning refers to the fine-tuning of network parameters based on RBM expression. Each program is run 10 times for each feature dimension, and their average result is taken, as shown in Fig. 6.

As can be seen from the above Figure, with the increase of the dimension of new features, the performance of RBM-expression classifier is lower than that of PCA expression, because PCA is still an effective method for linear feature extraction. However, relatively, the performance of RBM expression classifier is gradually improved, and the accuracy rate is more than 75%, which indicates that feature extraction based on RBM is feasible. The performance of RBM-fine-tuning classifier is higher than that of RBM-expression and PCA-expression, and the corresponding classifier performance does not change significantly. Although it fluctuates up and down, it maintains a relatively stable state, which verifies the stability and effectiveness of feature extraction based on RBM, and the reverse fine-tuning greatly improves the model accuracy. Through layer-by-layer unsupervised training and CD algorithm, the initial connection weights and bias of the first half of the network model can be obtained. After several tests and analysis, other initial set values of the network are shown in Table 4. Some set values in Table 2 are explained as follows: (1) The momentum learning rate in RBM is set as {0.5,0.9}, which means that the momentum learning rate of RBM is set as 0.5 at the beginning and increases to 0.9 when the reconstruction error is in a stationary decreasing state; (2) Two MSGD processes were carried out on the whole network, respectively in the process of RBM training and network fine-tuning, and the number of samples in small batches was set as 20. (3) The initial connection weights in RBM were derived from random numbers of normal distribution N (0,0.01), and the bias was initialized to 0. The initial parameters in Softmax classification

**Table 3 Sample data of the Wine Quality dataset.**

| 0 | Fixed acidity | Volatile acidity | Citric acid | Residual sugar | Chloriders | Free sulfur dioxide |
|---|---|---|---|---|---|---|
| 1 | 7.4 | 0.70 | 0.00 | 0.076 | 11.0 | 0.56 |
| 2 | 7.8 | 0.88 | 0.01 | 0.098 | 25.0 | 0.68 |
| 3 | 7.8 | 0.76 | 0.04 | 0.092 | 15.0 | 0.65 |
| 4 | 11.2 | 0.28 | 0.56 | 0.075 | 17.0 | 0.58 |
| 5 | 7.4 | 0.70 | 0.02 | 0.076 | 11.0 | 0.46 |
| 6 | 7.4 | 0.66 | 0.06 | 0.069 | 13.0 | 0.47 |
| 7 | 7.3 | 0.65 | 0.02 | 0.073 | 9.0 | 0.57 |

were derived from random numbers of normal distribution $0.05 * N(0,1)$. (4) In this article, the method using regularization item's aim is to avoid study appeared in the process of fitting, the corresponding coefficient is called regularization coefficient, after the objective function is the main approach and parameter of L1 norm or L2 norm, as to make the punishment for possible larger parameter, generally, have arisen because of the bias is unlikely to fitting, So we only use the regularization term for the join weights, and we don't use the bias term.

The reconstruction error can reflect the fitting degree of RBM to the training sample data to a certain extent. The number of nodes is set as (26,10,6,6) to obtain the reconstruction error. As can be seen from the Fig. 7, the initial value of reconstruction error of each RBM is higher than the final value of the previous RBM. The reason may be that when adding another layer of RBM, The connection weight of RBM is a random number from the normal distribution $N(0,0.01)$, so the initial reconstruction error may be at a high value. From the perspective of the reconstruction error of the RBM of the third layer, the reconstruction error of the RBM of the two layers does not decrease any more, and the reconstruction error of the RBM of the two layers basically coincides and tends to be stable, indicating that the RBM of the three layers can have a good fitting state for the training data. Therefore, the reconstruction error can be used to judge the number of hidden layers.

The number of hidden layers can be roughly judged from the reconstruction error, and then different number of hidden layers and nodes of hidden layers are selected to compare the influence of node number and layer number through the correct rate of Softmax classifier. Each group of experiments is run 10 times, and its average correct rate is obtained. The experimental results are shown in Fig. 8. Several layers of RBM in the legend refer to the feature extractor composed of several stacked RBMS. The number of RBM nodes in the first layer of the two-layer RBM is set as 26, and the number of RBM nodes in the first and second layers of the three-layer RBM is set as 26 and 10, respectively. The increasing number of nodes in the hidden layer of the abscissa heading refers to the change of the number of RBM nodes in the last layer.

As depicted in Fig. 8, the multi-layer RBM feature extractor significantly enhances the accuracy of the Softmax classifier. This improvement indicates that the feature data

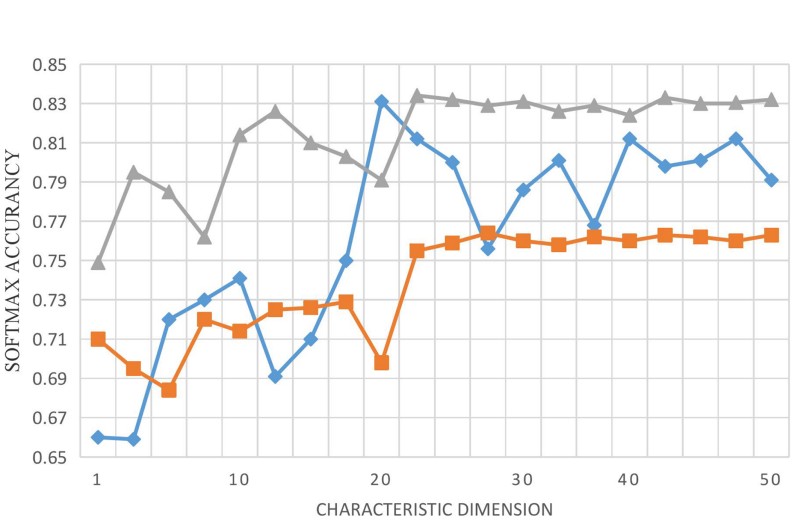

**Figure 6  Curve of accuracy *vs*. characteristic dimension.**

**Table 4  Network initialization set point table.**

| RBM | | Softmax | | Network tuning process | |
|---|---|---|---|---|---|
| CD iterations | 20 | Iterations | 300 | Number of fine adjustments | 200 |
| Rating | 0.1 | Rating | 0.25 | Rating | 0.5 |
| Momentum learning rate | {0.5,0.9} | Regularization coefficient | 0.004 | Momentum learning rate | 0.5 |
| Small batch learning rate | 20 | Initial parameters | $0.05 * N(0,10)$ | Regularization coefficient | 0.001 |
| Initial parameters | $N(0,0.01)$ | | | Small batch learning rate | 20 |

becomes more conducive to classification after undergoing layer-by-layer feature transformation. Additionally, the three-layer RBM demonstrates a relatively stable performance, suggesting that RBMs possess higher feature expression capabilities compared to single-layer RBMs. Thus, utilizing RBMs as the foundation for constructing the model in this study proves to be a feasible approach.

Based on the results presented in Fig. 8, it can be inferred that setting the number of iterations to 300 yields an accuracy of 78.3%. Both DBN-Softmax and the proposed method achieve higher accuracies than the traditional Softmax classifier. This observation confirms the effectiveness of extracting input data and using the new feature data for classification. The final experimental results are summarized in Table 5.

## Results of model comparsion

The parameters of the algorithm mainly include: population size $N$, the value is 20; Inertia weight $W$, generally 0.9; Learning factors $C_1$ and $C_2$, value 2.0; $r1$ and $r2$ are two random numbers in the interval [0,1]; The maximum velocity of the particle $V_{max}$ is 20; The termination condition is to meet the threshold, and the error accuracy is 0.001.

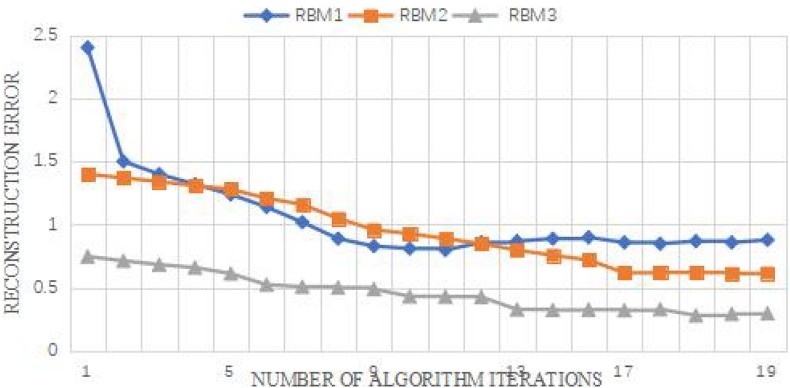

**Figure 7 Curve of accuracy *vs*. characteristic dimension.**

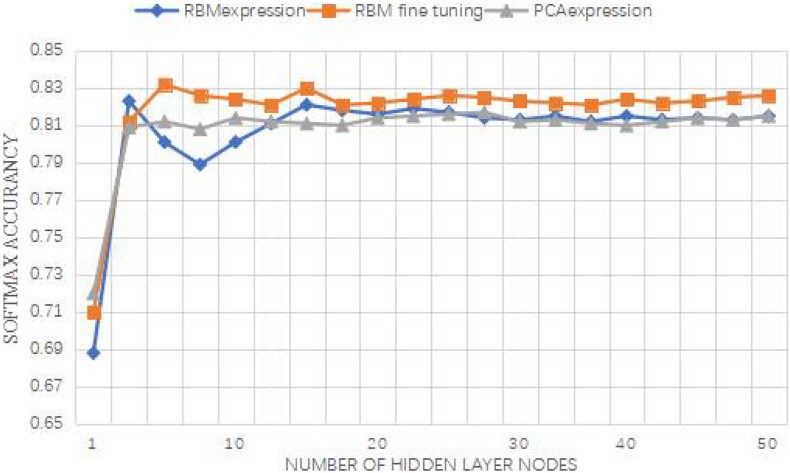

**Figure 8 Curve of accuracy *vs*. characteristic dimension.** For feature extractor with multi-layer RBM, the accuracy of Softmax classifier will be improved, that is, the performance of Softmax classifier will be improved with the increase of the number of network layers, indicating that feature data is more conducive to classification after feature transformation layer by layer.

From Figs. 9 and 10, it is evident that the improved PSO-BP algorithm outperforms the standard BP and standard PSO-BP algorithms in terms of convergence and prediction accuracy for both the Iris dataset and the WineQuality dataset.

For the Iris dataset, the iteration times of the BP algorithm are 69, while the standard PSO-BP algorithm requires 59 iterations, and the improved PSO-BP algorithm converges in just 50 iterations. Similarly, for the WineQuality dataset, the BP algorithm needs 69 iterations, the standard PSO-BP algorithm takes 61 iterations, and the improved PSO-BP algorithm converges in 52 iterations. The improved algorithm demonstrates a faster convergence speed compared to both the standard BP and standard PSO-BP algorithms.

Furthermore, all three algorithms, *i.e.*, the improved PSO-BP algorithm, the standard BP, and the standard PSO-BP algorithm, yield effective prediction outputs. However, by examining the error between the predicted values and the expected values, it becomes

Table 5 Comparative experimental results of RBM algorithm.

| Algorithm | AP% | AR% | Accuracy% |
|---|---|---|---|
| Softmax classifier | 78.7 | 73.75 | 73.33 |
| BP | 86.49 | 77.98 | 83.33 |
| DBN-Softmax | 84.90 | 83.94 | 85.00 |
| Our methods | 86.87 | 88.86 | 88.33 |

**Note:**
The accuracy of DBN-Softmax and the proposed method is higher than that of Softmax, which indicates that it is effective to extract the input data first and use the new feature data for classification.

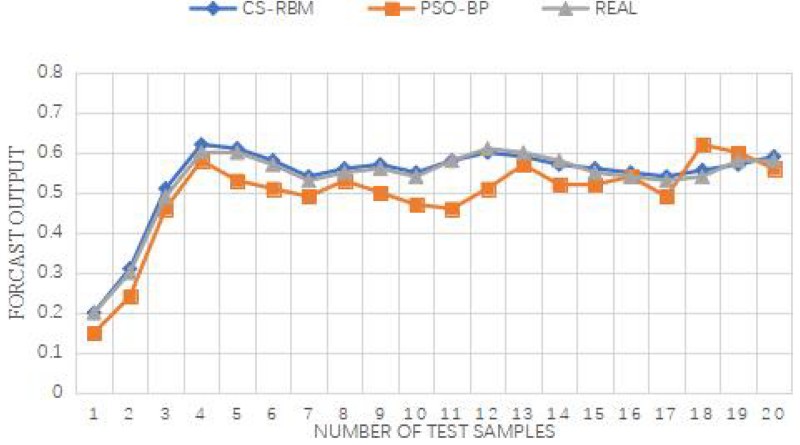

**Figure 9 Comparison between output and actual situation of Wine Quality datase.** The iteration times of the BP algorithm in the WineQuality dataset is 69, the iteration times of the standard the PSO-BP algorithm is 61, and the iteration time of the improved PSO-BP algorithm is 52.

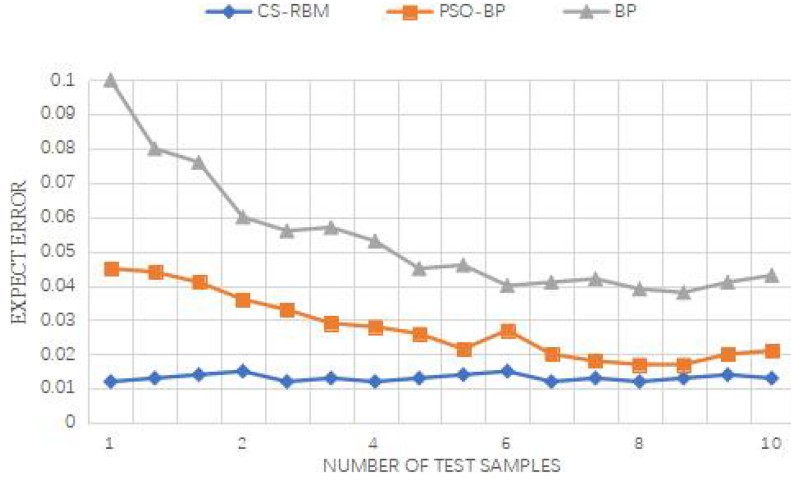

**Figure 10 Iris data set error between the predicted value and the expected value.**

evident that the CS-RBM (improved PSO-BP) algorithm achieves better results than the PSO-BP algorithm. Overall, the simulation experiments demonstrate the superior performance of the improved PSO-BP algorithm in terms of convergence speed and prediction accuracy for both the Iris dataset and the WineQuality dataset, making it a promising approach for various prediction tasks.

## Index analysis

By PSO and BP neural network algorithm, the use of information project build system of performance indicators, with 12 colleges and universities campus cloud desktop system construction project as an example, from the user's satisfaction, project profit, quality, application of four aspects to obtain the data of the construction of the project, and the data were normalized processing, set the PSO and BP neural network initialization parameters. The evaluation results are predicted by the algorithm. According to the construction content of the information project education management system in this article, the construction goal and performance index system of the university cloud desktop construction project are formed by using user satisfaction, project revenue, project quality and performance index system in project application. There are six first-level indicators and 27 second-level indicators. The university desktop cloud system project needs to collect data from 27 secondary indicators including user satisfaction, project revenue, project quality, and project application. There are two main ways to collect data. One is to collect log data in the system project. The other is to collect the project data of desktop cloud systems of 12 universities by questionnaire. In addition to the data collection of 27 second-level indicators, it is also necessary to investigate the overall satisfaction of desktop cloud system projects in 12 universities. The satisfaction is formed to compare and analyze the expected value predicted by PSO-BP neural network and verify the effectiveness of PSO-BP neural network model. Table 6 shows the results of a questionnaire administered to 12 colleges and universities. A total of 30 administrative staff, 70 full-time teachers and 500 students completed a satisfaction assessment.

According to the above statistical data, the user satisfaction of A university is 97%. In the same way, the satisfaction of other university users is respectively counted, as shown in Table 7.

According to obtain data from college cloud desktop system in the project, we to normalization of data indicators, indicators describe the content of the different, the range of values is not the same, the index data processing to index of the maximum and the minimum differential mathematical formula calculation, calculation results between the [0, 1], need to normalized processing of raw data. The formula for normalization is as follows:

$$y_i = \frac{x_i - \min(x)}{\max(x) - \min(x)} \tag{16}$$

where $y_i$ is the standard value of the index; $x_i$ is the measured value of an index; $\max(x)$ is the maximum value of an indicator; $\min(x)$ is the minimum value of the index. The normalization formula above is used to process the data obtained from the university cloud

**Table 6 Statistical table of university satisfaction survey.**

| Investigation type | Very satisfied | Satisfied | Middle satisfied | General | Not satisfied |
|---|---|---|---|---|---|
| Administrative personnel | 5 | 19 | 3 | 2 | 1 |
| Teacher | 2 | 48 | 16 | 1 | 3 |
| Student | 109 | 317 | 64 | 5 | 5 |

desktop project. $x_1$–$x_{12}$ is the input eigenvalue of the second-level indicator, and $A$-$K$ is the cloud desktop system project of 12 universities. After the above secondary data index collection, normalization processing, neural network input layer, hidden layer, output layer neurons determination, inertia weight and learning factor initial value determination, as well as evaluation criteria. Input the data set, and calculate the expected value through the CS-RBM neural network. The performance comprehensive evaluation value of the university cloud desktop system, the predicted value and evaluation value are shown in Table 8.

Through the comparison with the survey data, as shown in Table 8, it can be seen that it is basically consistent with the survey comprehensive evaluation data, and the result is consistent with the expected value. The coincidence rate is 98%. Therefore, it can be seen that the improved neural network established in this article is reasonable and effective.

## Discussion

The present study focused on the evaluation of students' innovative ability in higher education, incorporating their psychological cognition level into the assessment process. By utilizing the DELPHI method to quantify qualitative data and adopting the IPSO-LSTM model for optimization, the researchers achieved promising results with high accuracy and improved model convergence speed. Additionally, the analysis of students with excellent innovation abilities revealed the significant contribution of psychological indicators to their innovative prowess, emphasizing the importance of nurturing students' psychological cognition level for fostering innovation capabilities.

The implications of these findings extend beyond the scope of this specific study. Firstly, the integration of psychological factors into the evaluation of innovation ability opens up new avenues for research in higher education. It highlights the multifaceted nature of innovation, where psychological attributes and cognitive abilities play a crucial role alongside traditional skill-based assessments. This can inform curriculum development, teaching methodologies, and the design of innovative education programs. The IPSO-LSTM model's success in optimizing the evaluation process showcases the potential of hybrid optimization techniques in machine learning applications. By combining particle swarm optimization with long short-term memory networks, the IPSO-LSTM model provides a more efficient and accurate approach to evaluating students' innovation abilities. This advancement can find applications in other domains that require precise and timely decision-making, such as predictive analytics, medical diagnosis, and financial forecasting.

**Table 7 Statistical table of college satisfaction.**

| Code | X1 | X2 | X3 | X4 | X5 | X6 | X7 | X8 | X9 | X10 | X11 | X12 |
|---|---|---|---|---|---|---|---|---|---|---|---|---|
| Satisfaction | 97% | 85% | 89% | 94% | 83% | 90% | 97% | 98% | 96% | 93% | 93% | 88% |

**Table 8 Statistical table of satisfaction evaluation value and prediction value of all colleges and universities.**

| Code | X1 | X2 | X3 | X4 | X5 | X6 | X7 | X8 | X9 | X10 | X11 | X12 |
|---|---|---|---|---|---|---|---|---|---|---|---|---|
| Satisfaction | 97% | 85% | 89% | 94% | 83% | 90% | 97% | 98% | 96% | 93% | 93% | 88% |
| Prediction | 96% | 84% | 89% | 92% | 82% | 90% | 96% | 98% | 94% | 93% | 94% | 89% |

Moreover, the characteristic contribution analysis sheds light on the complex relationship between psychological factors and innovation capability. This understanding can inform targeted interventions and educational strategies to cultivate students' innovative mindsets and creative thinking. Universities and educational institutions can use this insight to implement more effective mental health education and creativity cultivation programs, ultimately fostering a more innovative and resilient student population.

## CONCLUSION

Performance of information management in colleges and universities is increasing in demand, based on the existing informatization project based on the theory of education management method based on the analysis to evaluate the project anticipated target, analyze the target content, determine its performance indicators, determine the logic level of performance indicators, using the index of effective tools and methods to collect data, Through the improved CS-RBM neural network algorithm to calculate the data results, combined with the user satisfaction data obtained from the survey, finally form the education management conclusion of the project. Experimental results show that the accuracy of CS-RBM algorithm is 15.05% higher than that of the same type algorithm on the public data set, and the prediction consistency reaches 98.8%. The effectiveness of the network established in this article is evident, as demonstrated by the impressive results achieved in evaluating students' innovation ability based on their psychological cognition level. However, it is acknowledged that the early work in the evaluation process, particularly in data acquisition and normalization, can be cumbersome and time-consuming. Therefore, it is important for future research to focus on improving the data acquisition and normalization methods for performance indicators.

## ACKNOWLEDGEMENTS

The author would like to thank the anonymous reviewers who contributed valuable comments to this article.

### Funding
This work was supported by the 2021 Henan Higher Education Teaching Reform Research and the Practice Project of the Education Department Of Henan Province, "Construction and Practice of the Curriculum System for Connecting Industrial Chain Specialty Groups in Higher Vocational Colleges—Taking Financial Technology Application Specialty Groups as an Example" (No. 2021SJGLX723). The funders had no role in study design, data collection and analysis, decision to publish, or preparation of the manuscript.

### Grant Disclosures
The following grant information was disclosed by the authors:
2021 Henan Higher Education Teaching Reform Research and Practice Project of the Education Department of Henan Province: 2021SJGLX723.

### Competing Interests
The authors declare that they have no competing interests.

### Author Contributions
- Huifang Guo conceived and designed the experiments, performed the experiments, analyzed the data, performed the computation work, prepared figures and/or tables, authored or reviewed drafts of the article, and approved the final draft.

### Data Availability
The data set is available at Zenodo: Giovanni Salucci. (2023). Business schools dataset (v.1) [Data set]. Zenodo. https://doi.org/10.5281/zenodo.7863501.
The code for this article are available in the Supplemental Files.

### Supplemental Information
Supplemental information for this article can be found online at http://dx.doi.org/10.7717/peerj-cs.1575#supplemental-information.

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
