# Peer review of "The evaluation of university management performance using the CS-RBM algorithm"

_PeerJ Computer Science, doi:10.7717/peerj-cs.1575_

## Round 0.1 · original submission · Major Revisions

Dear authors,

Two experts in the field have reviewed your manuscript, and you will see that they have could of comments for improvement of the article. Therefore, you are requested to update the article in light of those comments and resubmit for re-evaluation.

Please also improve the title of your article as currently, it seems too long and a bit confusing.

Please also improve the abstract to include the findings of your research.
and the language quality.

**Language Note:** The Academic Editor has identified that the English language must be improved. PeerJ can provide language editing services - please contact us at copyediting@peerj.com for pricing (be sure to provide your manuscript number and title). Alternatively, you should make your own arrangements to improve the language quality and provide details in your response letter. – PeerJ Staff

Reviewer 1 ·

Basic reporting

Details are available in the additional comment section.

Experimental design

The research question is relevant and meaningful.

Validity of the findings

The data on which the conclusions are based made available in an acceptable discipline-specific repository. The data is robust, statistically sound, and controlled.

Additional comments

This article employs a deep learning algorithm to conduct quality assessment of higher education, which holds significant application and dissemination potential. Overall , the paper seems good and easy to follow, However, there are still areas that require improvement in this study.

1.Within the Related Works section, the author presents a range of studies on university performance management aided by information technology. How can the author pursue further innovative research based on the aforementioned studies?
2.Author need to contextualize the findings in the literature, and need to be explicit about the added value of your study towards that literature.
3.In the training process of the model, a two-layer RBMs deep learning network is utilized. Please include a schematic diagram to visually illustrate the network structure and better convey the research concepts in this section.
4.While addressing the model-solving process, the author proposes a modular deep learning model based on RBMs for evaluating university performance data. Please provide additional details and specific steps within this section.
5.Discussion section needs to be a coherent and cohesive set of arguments that take us beyond this study in particular, and help us see the relevance of what the authors have proposed.
6.During the model construction phase, the author introduces the BGD (Batch Gradient Descent) algorithm but opts to use MBGD (Mini-Batch Gradient Descent) instead. Please request the author to provide an explanation for this choice.
7.In the Conclusion section, the author outlines the main contributions of this article. Please request the author to discuss any shortcomings or future plans for further development in their work.
8.The authors must have their work reviewed by a proper translation/reviewing service before submission; only then can a proper review be performed.
9.Most sentences contain grammatical and/or spelling mistakes or are not complete sentences.

Cite this review as

Reviewer 2 ·

Basic reporting

This paper establishes a novel multi-classification model that integrates multi-layer RBMs for feature extraction in deep learning and a Softmax classifier. The model is employed to predict the performance evaluation of colleges and universities, yielding more precise evaluation outcomes. While this paper exhibits certain innovative aspects, there are still specific details that require further explanation or improvement.

(1) In the introduction section of this paper, the author introduced numerous studies on university performance evaluation without undertaking an analysis of the inherent limitations of these works. Please provide an explanation for this omission.
(2) Within the Related Works section, the author explicated a range of endeavors concerning university performance management leveraging information technology. How can the author embark upon further innovative endeavors based on the aforementioned research?
(3) While constructing the network architecture of the model, the author augmented a stack-based limited Boltzmann machine upon the foundation of the DBN model. Please elucidate the background of the DBN model beforehand.
(4) During the model-solving process, the author posits a modular deep learning model based on RBMs for the evaluation of university performance data. Kindly supplement the specific steps in this section.
(5) Within the Introduction, the author contends that digital technology has become an indispensable facet of contemporary life, owing to the swift advancement of computer networks and information systems. What are the advantages of utilizing digital technology to quantitatively measure teacher performance evaluation indicators in a scientifically rigorous manner?
(6) In the analysis of model output results, three samples of input data and the corresponding output values of RBMs at each layer are randomly selected as the dataset for assessing the performance of the RBMs' feature extractor. These values are then depicted in a line chart. Could the author provide an explanation for this outcome?
(7) The author evaluates the proposed model based on average precision, average recall, and prediction accuracy, and illustrates a comparison of evaluation indicators for each algorithm. Please add a conclusion that can be drawn from this comparative analysis of the indicators.

Experimental design

Details can be found in the basic reporting.

Validity of the findings

Details can be found in the basic reporting.

Cite this review as

---

## Round 0.2 · accepted · Accept

Based on the input from the experts, your paper is being recommended for publication. Congratulations!!!

Reviewer 1 ·

Basic reporting

The article fulfills all the requirements.

Experimental design

Research question is relevant and through investigations performed up to the standard.

Validity of the findings

All the data provided in the paper is robust and statistically sound. Conclusion is fully linked to the original research question.

Cite this review as

Reviewer 2 ·

Basic reporting

The authors have addressed all the observations.

Experimental design

.

Validity of the findings

.

Additional comments

.

Cite this review as